environmental chemistry/green chemistry/chemical engineering

desulfurization, benzotiophenic compounds, liquid fuel, extraction, diamine-terminated oligomeric polyethylene glycol, 4,7,10-trioxatridecane-1,13-diamine

**Author for correspondence:**
Effat Kianpour
e-mail: e.kianpour@umz.ac.ir; efat.kianpour@gmail.com

# Extractive desulfurization of liquid fuel using diamine-terminated polyethylene glycol as a very low vapour pressure and green molecular solvent

Fatemeh Rafiei Moghadam[1], Effat Kianpour[3], Saeid Azizian[1], Meysam Yarie[2] and Mohammad Ali Zolfigol[2]

[1]Department of Physical Chemistry, Faculty of Chemistry, and [2]Department of Organic Chemistry, Faculty of Chemistry, Bu-Ali Sina University, Hamedan 65167, Iran
[3]Department of Physical Chemistry, Faculty of Chemistry, University of Mazandaran, PO Box 47416-9544, Babolsar, Iran

EK, 0000-0003-2808-8115

Removal of sulfur compounds from liquid fuel is one of the important issues in the field of energy and environment. Among the available methods, extractive desulfurization (EDS) is of great interest due to its convenient operating conditions. In this study, EDS performance of 4,7,10-trioxatridecane-1,13-diamine (TTD), a very low vapour pressure diamine-terminated oligomeric polyethylene glycol (PEG), was studied. Effect of the influencing factors, as well as multiple extraction, mutual solubility, reusability and regeneration of TTD were investigated. Results showed that the TTD/fuel volume ratio of 0.5 could extract benzothiophene, dibenzothiophene and dimethyl dibenzothiophene with the efficiencies 67%, 74% and 53%, respectively, in less than 1 min at ambient temperature. The distribution coefficient ($K_N$) value for removal of dibenzothiophene by TTD was 3.66 higher than that of PEG, and it is similar to $K_N$ values (approx. 4) for polyethylene glycol dimethyl ether (as a modified PEG) and Lewis acid-containing ionic liquids. It was observed that spent TTD after five cycles could be regenerated using the back-extraction method. Also, deep EDS was achievable after three times extraction using fresh TTD. Finally, the extraction mechanism was studied using [1]H-NMR. These observations, as well as very low vapour pressure and insignificant dependency of TTD on the initial S-concentration of fuel and temperature, make this extractant to be introduced as a valuable option for green and effective EDS.

# 1. Introduction

Despite the emergence of new energies, fossil fuel still plays an important role in supplying energy in different aspects of human life. The major drawback of using these fuels, despite their high energy density, is environmental pollution problems. As a result of fuel combustion, sulfur oxides ($SO_x$) are produced and enter the atmosphere. This leads to air pollution, acid rain, depletion of the ozone layer and other dangerous environmental impacts [1]. Therefore, worldwide standards and regulations have been established to reduce sulfur compounds in the fuel to approximately or even lower than 10 ppm [2]. Moreover, sulfur compounds in the fuel cause corrosion of plants and inactivation of industrial catalysts [1]. So, the removal of sulfur compounds from liquid fuels is one of the concerns of researchers and industrials. The catalytic hydrodesulfurization (HDS), which requires harsh and high-cost operating conditions (temperature: 300–400°C, pressure: 2–10 MPa of $H_2$ and expensive catalysts) [3], is the traditionally used method in the industry. Although 70% of S-compounds of liquid fuel are benzothiophene (BT), dibenzothiophene (DBT) and their derivatives [4]; however, these cyclic sulfur compounds have limited activity under the usual conditions of HDS method. Therefore, it is not possible to achieve deep desulfurization (S-content less than 10 ppm) by HDS [2]. Oxidation, adsorption, extraction and biodesulfurization are some of the alternative or complementary methods for HDS [5]. Among these, the extractive desulfurization has received special attention due to its convenient operating conditions (ambient temperature and pressure), easy separation of refined fuel and solvent and low cost [6–13]. Moreover, extraction is not a destructive method, i.e. does not change the quality of liquid fuels and chemical structure of valuable S-compounds in the fuel (which can be used as starting materials in different industries) [8,10]. Because the extraction process is significantly influenced by the type of extractant, extractive desulfurization (EDS) performance of different solvent has been investigated up to now [6–14].

Initial reports on using molecular solvents, ordinary liquids such as water which are predominantly made of electrically neutral molecules (non-polar or polar molecules but not ionic molecules), in EDS come from nitrogen-containing solvents such as N-methyl-2-pyrrolidone (NMP), 1,3-dimethyl-2-imidazolidinone (DMI) and dimethylformamide (DMF), as found in the literature [13]. Despite the good EDS performance, high volatility has become a major obstacle to the application of these solvents on the industrial scale [8]. To avoid the disadvantage of volatile molecular solvent, ionic liquids have been introduced as green solvents for EDS since 2001 [15], because of low vapour pressure. But the high viscosity and high cost have challenged the practical application of them in EDS [10]. Recently, polyethylene glycol (PEG-200), with the vapour pressure less than 0.01 hPa, was introduced as a green, cheap and efficient extractant for desulfurization [10]. Then, Gao *et al.* showed that the modification of PEG, via replacement of terminal –OH groups with –$OCH_3$ groups, can improve the value of $K_N$ and therefore the extraction performance of this extractant [8].

4,7,10-Trioxatridecane-1,13-diamine (TTD) is a commercially available diamino-terminated oligomeric PEG (extractive desulfurization has received special attention due to its convenient operating conditions at ambient temperature [16–18]), which is used as a curing agent (hardener) in manufacturing of activated resin/beads, as a reagent includes cross-linker containing PEG in pharmaceuticals, in hair salons, etc. [16,19–21]. Moreover, it has a very low vapour pressure (less than 0.001 hPa, even less than that of PEG and also polyethylene glycol dimethyl ether (NHD, as a modified PEG)) [8,22]. TTD can be considered as an oligomeric PEG in which its terminal –OH groups are replaced with –$NH_2$ groups. Based on that mentioned and in continuation of our interest in EDS studies [9–12], the performance of TTD in the extraction of refractory S-compound, i.e. BT, DBT and dimethyl dibenzothiophene (DMDBT), from the liquid fuel was investigated in this work, and it was found that this molecular solvent can be introduced as a non-volatile nitrogen-containing molecular extractant with the good performance and results in EDS process.

# 2. Material and methods

## 2.1. Materials

4,7,10-Trioxatridecane-1,13-diamine (purity of 98%), heptane (purity of 99%) and pentane (purity of 99%) were purchased from Merck Company. Also, benzothiophene, dibenzothiophene and dimethyl dibenzothiophene (purities of 98%, from Sigma-Aldrich Co.) were used in this work.

## 2.2. Extractive desulfurization process

To prepare the model liquid fuel at a specified concentration, the desired sulfur compound, i.e. BT, DBT or DMDBT, was dissolved in heptane. The procedure of all tests to investigate influencing factors on EDS by TTD is as follows: the glass bottle containing a binary mixture of extractant and model fuel containing DBT, with a specified volume ratio of TTD/fuel, was placed on a stirrer at the desired temperature. It should be noted that the detailed information about the concentration of DBT, temperature and volume ratio of TTD/fuel in each experiment have been presented in the figure captions. After magnetically mixing the two phases, the extraction process was stopped at the desired time. Desired times mean 5, 25, 40, 60, 90 and 180 s in kinetic tests and 20 min in equilibrium experiments. After stopping the stirring and complete phase separation, the fuel phase was sampled to measure the amount of residual sulfur compound. It should be noted that the sulfur compound concentration in the fuel was measured by the UV/Vis spectrophotometer (PG Instrument Ltd, T-80) at the corresponding $\lambda_{\max}$ [23]. Then, to evaluate the desulfurization performance of extractant, extraction efficiency (EE (%)) and distribution coefficient ($K_N$) were calculated using equations (2.1) and (2.2)

$$EE(\%) = \frac{C_0 - C_t}{C_0} \times 100 \tag{2.1}$$

and

$$K_N = \frac{C_0 - C_t}{C_t} \times \frac{m_{\text{fuel}}}{m_{\text{TTD}}}, \tag{2.2}$$

where $C_0$ and $C_t$ are initial and at any time concentrations of sulfur compound (ppm) and $m_{\text{fuel}}$ and $m_{\text{TTD}}$ represent the fuel and extractant mass, respectively.

To investigate the EDS performance of TTD for other refractory S-compounds, a binary mixture of TTD and model fuel containing approximately 500 ppm BT, DBT or DMDBT, with a volume ratio of 0.5 (TTD/fuel), was a stirred room temperature for 20 min. After that, the fuel phase was sampled to measure the residual amount of each sulfur compound.

To achieve deep desulfurization, the glass bottle containing a binary mixture of TTD and model fuel containing approximately 500 ppm DBT, with a volume ratio of 0.5 (TTD/fuel), was stirred at room temperature for 20 min. After separating the TTD phase, the treated fuel was mixed with the fresh extractant in the next stage. This protocol was repeated again and again until the goal (fuel with sulfur concentration less than 10 ppm) was achieved.

To check the mutual solubility of the fuel and the extractant, pre-dried TTD (under vacuum conditions and temperature 50°C) was exposed to heptane for 20 min and the mixture was magnetically stirred. After stopping and separation of two phases, the UV/Vis spectrum of heptane, saturated with TTD, was recorded. Also, to investigate the solubility of the fuel in the extractant, the specific weight of TTD, saturated by heptane, was dried for 24 h under vacuum at 50°C and its final weight was recorded. Fuel solubility in TTD in terms of mass percentage (wt%) was evaluated using $(W_1 - W_2)/W_1 \times 100$, in which $W_1$ and $W_2$ are the weight of saturated TTD by heptane and dried extractant, respectively.

To re-use the TTD in the extraction process, the used extractant in the first cycle of EDS was separated and mixed with the fresh model fuel containing approximately 500 ppm DBT with a volume ratio of 0.5 (TTD/fuel) at room temperature for 20 min. Exposing the consumed TTD with fresh fuel was iterated until the saturation of extractant. For regeneration of TTD, spent extractant after five cycles re-using was mixed with pentane with the volume ratio pentane/TTD = 2.5 (named: Reg. a) or 5 (named: Reg. b) which are equal to 0.4 and 0.2 TTD/pentane volume ratio, respectively. After 20 min stirring and phase separation, regenerated TTD was exposed to fresh fuel containing approximately 500 ppm DBT for 20 min at room temperature. Finally, sampling was done to measure the amount of residual sulfur compound and the calculation of the efficiency of this regeneration method. It should be noted that the details of each experiment have been presented in the figure captions.

To study the extraction mechanism, [1]H-NMR (Avance III-400, Bruker) was employed using CDCl$_3$ as the solvent.

## 3. Results and discussion

To disclose the overall performance of the TTD, the influence of factors: the extractant/fuel volume ratio, process time, temperature and initial S-compound concentration in the fuel on the extractive

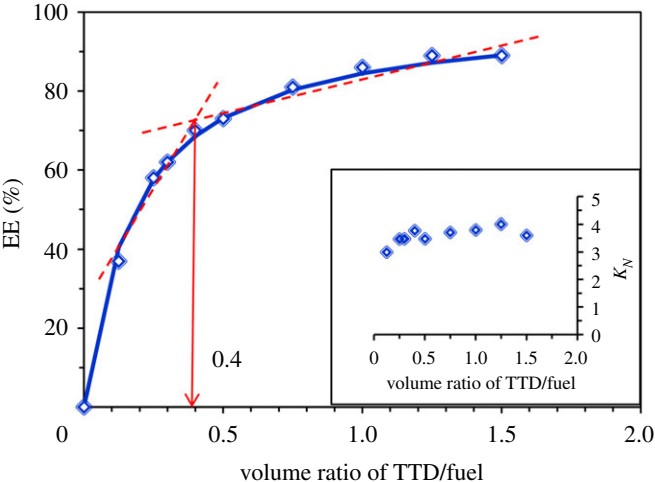

**Figure 1.** Effect of TTD/fuel volume ratio on EE (%) and $K_N$ (room temperature, sampling after 20 min stirring, initial concentration of DBT approximately 500 ppm).

desulfurization process as well as the effect of S-compound type, multiple extractions, mutual solubility, reusability, regeneration of extractant and the extraction mechanism was investigated in this work.

## 3.1. Influencing factors on extractive desulfurization by 4,7,10-trioxatridecane-1,13-diamine

Fulfilment of extractive desulfurization with minimum solvent consumption is the most favourable from the environment and economic points of view. Optimization of the extractant/fuel volume ratio is a way to achieve this goal. Therefore, the effect of TTD/fuel volume ratio on the removal of DBT (as a typical refractory S-compound) was investigated via exposing a constant volume of liquid fuel (with the concentration of 500 ppm) to the different amounts of extractant during several experiments for 20 min. Results (figure 1) showed that the extraction efficiency enhanced with the increase of TTD/ fuel volume ratio. But the incremental trend of EE did not have the same slope in the whole range and, as can be seen, the TTD/fuel volume ratio of 0.4 can be considered as the intersection of two zones with different slopes so that after this volume ratio the EE has slowly increased. Therefore this volume ratio, which shows the EE 70% for DBT removal, can be considered as an optimum point. To make sure of choosing an optimum condition, the TTD/fuel volume ratio of 0.5 was used in the following investigations. Based on the TTD and heptane densities, this volume ratio, which removes 74% of DBT from fuel, is equal to 0.74 TTD/fuel mass ratio. In the following, the value of $K_N$ was calculated for different volume ratios (inset of figure 1). It was observed that $K_N$ (calculated using equation (2.2)) did not significantly change with the variation of volume ratio, and the mean value of this thermodynamic quantity is 3.66 in (ppm S)$_{TTD}$/(ppm S)$_{fuel}$. The value of $K_N$ also can be calculated via fitting the obtained experimental data for EE versus volume ratio with the following equation, as described in our previous works [11,12]:

$$\text{EE } (\%) = \left( \frac{K_N(V_{TTD}/V_{Fuel})}{K_N(V_{TTD}/V_{Fuel}) + 1} \right) \times 100 \,, \qquad (3.1)$$

where $V_{fuel}$ and $V_{TTD}$ represent the fuel and extractant volume, respectively. The solid line in figure 1 represents the result of this fitting, which is completely consistent with experimental data. It was observed that the value of $K_N$ obtained by this method is 5.45 in (molar concentration of S)$_{TTD}$/(molar concentration of S)$_{fuel}$, which is equal to 3.43 in (ppm S)$_{TTD}$/(ppm S)$_{fuel}$, and it is in agreement with the mean value of experimentally obtained $K_N$ (3.66). Recently, Kianpour & Azizian reported that the PEG-200/fuel volume ratio of 1 can remove 75% of DBT with the $K_N$ value of 1.8 [10]. Also, Gao *et al.* showed that polyethylene glycol dimethyl ether (NHD, as a modified PEG) can approximately extract 80% of DBT at extractant/fuel mass ratio of 1 with the $K_N$ value of 3.99 which is similar to that of reported values for Lewis acid-containing ionic liquids such as [Bmim][AlCl$_4$] (approx. 4.09) and NMP-SnCl$_2$ (approx. 4.06) [8]. Based on the above discussion, it can be concluded that TTD, a diamino-terminated oligomeric polyethylene glycol, is a more effective extractant for the EDS process in comparison with PEG and modified PEG (via replacement of terminal –OH groups with –OCH$_3$ groups).

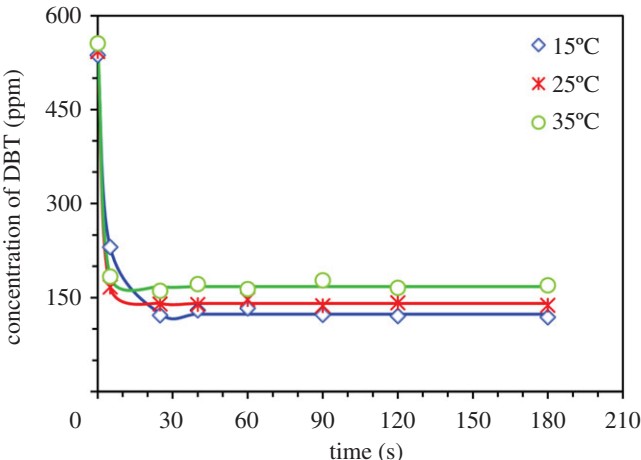

**Figure 2.** Effect of temperature on concentration variation of DBT with the time during the extraction by TTD (TTD/fuel volume ratio 0.5, initial concentration of DBT approximately 500 ppm).

The time dependence of DBT extraction by TTD is shown in figure 2. It can be seen that the equilibrium condition was reached very quickly, within less than 1 min, while the equilibrium time for PEG-200 was 2 min [10]. Indeed, the lower viscosity of TTD (13–14 cP at 20°C) in comparison with that of PEG-200 (60 cP at 20°C) leads to faster mass transfer during the extraction process [22]. In the equilibrium studies, fuel phase sampling was done after 20 min extraction to assure reaching the equilibrium condition.

As mentioned in introduction, extractive desulfurization has received special attention due to its convenient operating conditions including ambient temperature. Therefore taking into account 10°C fluctuations in ambient temperature, temperature range 15–35°C was chosen to investigate the effect of this parameter on the EDS. Therefore, the time variation of DBT concentration was recorded at 15, 25 and 35°C to investigate the effect of temperature on the EDS process by TTD. The results (figure 2) show that the temperature dependence of the extractive performance of TTD is very low. This observation, which can be related to the low viscosity of TTD, is economically desirable. Because of very low-temperature dependency of EDS using TTD the room temperature can be chosen as the optimum condition for the extraction process. It was observed that the calculated EE decreased from 77 to 70% with the increase of temperature, which can be attributed to the evaporation of heptane. Moreover, the experimental data were best fitted with the first-order kinetic model: $C_t = ((C_0 - C_e)e^{-kt}) + C_e$, where the $C_0$, $C_e$ and $C_t$ are initial, equilibrium and at any time concentrations, respectively and the values of 0.27, 0.55 and 0.64 s$^{-1}$ were obtained for the rate constant ($k$) of DBT extraction by TTD at 15°C, 25°C and 35°C, respectively. These are much higher than the previously reported values of $k$ for PEG-200 at the same temperatures [10].

Since influents with different sulfur concentrations are injected into the desulfurization units of refineries, the effect of initial S-compound concentrations on the EDS performance of TTD has been investigated in the following. The initial concentrations were chosen based on common concentrations investigated in this type of research [24,25]. Results (figure 3) show that the decrease of the EE with the increase of DBT concentration is not considerable. Also, the investigation of $K_N$ values indicates the low concentration dependency of this quantity so that the $K_N$ values only decrease from 3.78 to 3.01 with the increase of DBT concentration from 300 to 1200 ppm. Therefore, it can be concluded that TTD has good performance for EDS of liquid fuels with different S-compound concentration.

## 3.2. EDS performance of 4,7,10-trioxatridecane-1,13-diamine for other refractory S-compounds

Because of the diversity of refractory S-compounds in liquid fuel, the effect of benzotiophenic compound type on the desulfurization performance of extractant should be investigated, too. For this purpose, the value of EE (%) and $K_N$ for the removal of BT, DBT and DMDBT by TTD was compared after 20 min. As presented in table 1, the variation of EE (%) and $K_N$ values with the change of S-compound type follows the order of DBT > BT > DMDBT with the value of $K_N$: 3.66, 2.57 and 1.42 in (ppm S)$_{TTD}$/(ppm S)$_{fuel}$ and EE: 74%, 67% and 53%, respectively. The higher electron density, as well as the lower steric hindering, of

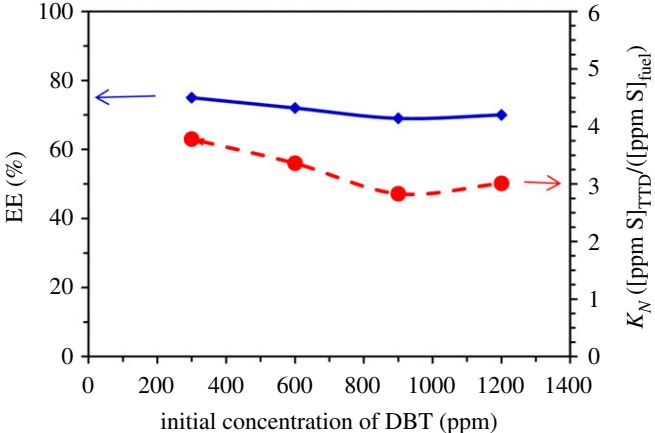

**Figure 3.** Effect of the initial concentration of DBT on $K_N$ and EE (%) (room temperature, stirring time of 20 min, TTD/fuel volume ratio 0.5).

**Table 1.** Effect of refractory sulfur compound type on EDS performance of TTD (room temperature, stirring time of 20 min, extractant/fuel volume ratio 0.5, initial concentration of sulfur compound approximately 500 ppm).

| S-compound | EE (%) | $K_N$ |
|---|---|---|
| DBT | 74 | 3.66 |
| BT | 67 | 2.57 |
| DMDBT | 53 | 1.42 |

sulfur atom of DBT in comparison with that of BT and DMDBT, respectively, may have led to the more efficient interaction and therefore extraction of DBT by TTD [10].

## 3.3. Multiple extraction

Deep EDS cannot achieve via the single-step extraction method with the extractant having the $K_N$ value below 10, and multiple extractions with the fresh extractant are the solution to this problem [26]. The performance of TTD (with the volume ratio of extractant/fuel = 0.5) during the multiple extractions of DBT from the liquid fuel is depicted in figure 4 as residual concentration of DBT. It was observed that the concentration of DBT reduced from 500 to 20 ppm by the second cycle and deep desulfurization was achievable after three times extraction with the fresh TTD. Although the same results were observed for PEG [10], it should be noted that the volume ratio of extractant to fuel was 1 in that case. Moreover, these results show that TTD is more efficient during multiple extraction processes in comparison with reported work about ionic liquids which need at least four times extraction to attain deep desulfurization [9,11,12].

## 3.4. Reusability and regeneration of 4,7,10-trioxatridecane-1,13-diamine

From the economic point of view, the reusability (without regeneration) and also the regeneration of the spent extractant are the important issues in the extraction-based process. The performance of TTD during the re-using process (without regeneration) for EDS of DBT from liquid fuel is presented in figure 5. It can be seen that the extraction capacity of used TTD for the removal of DBT from fresh fuel decreased with the increase of cycle number. From the results, it is observed that the EE of TTD decreased from 74 to 51% in the first step of reuse and after five times the extractant almost becomes saturated (EE of 9%) and should be regenerated. Different methods including back extraction, distillation and water dilution are reported [9] for regeneration. Although most of them do not seem economically and environmentally desirable, back extraction using light alkanes (C2–C5) as re-extraction media would be interesting [15]. Because these solvents have low boiling points, they can be easily recovered and separated from these sulfur compounds by evaporation. Then the organic sulfur could be converted into elemental sulfur

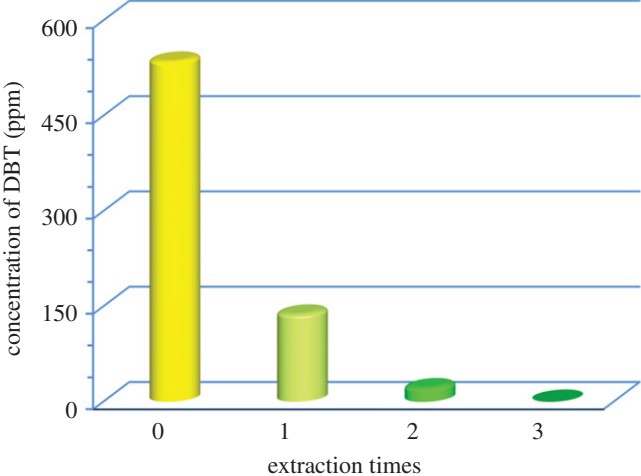

**Figure 4.** Concentration variation of DBT during multistep extractions (room temperature, stirring time of 20 min, TTD/fuel volume ratio 0.5, initial concentration of DBT approximately 500 ppm).

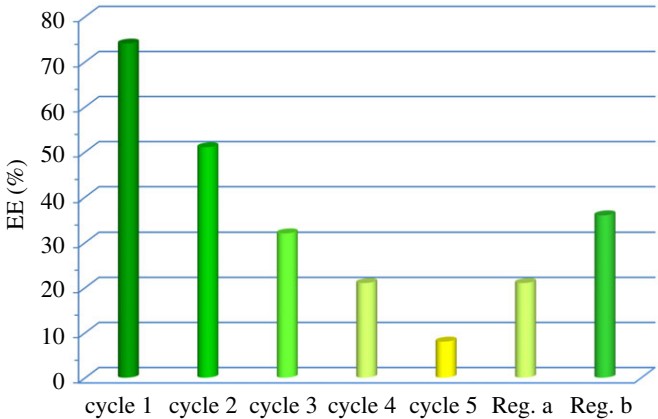

**Figure 5.** The extraction efficiency of DBT during several times reuse (cycles 1–5) and after regeneration (Reg. a and b) of TTD (room temperature, stirring time of 20 min, TTD/fuel volume ratio 0.5, initial concentration of DBT approximately 500 ppm).

by the common Claus process. In this work, pentane was used as a re-extraction medium for the regeneration of spent TTD after five times reuse in the EDS process. As can be seen from figure 5 (Reg. a and b), EE of DBT by TTD increased after regeneration of spent extractant using pentane. Moreover, the extraction ability of TTD increased from 21 to 36% with the increase of the TTD/ pentane volume ratio from 2.5 to 5. Therefore, it can be expected that the complete regeneration of TTD can be achieved by using more pentane. It should be noted that this solvent can be easily recovered by evaporation. It is to be noted that the extraction ability of DBT by TTD after regenerations (Reg. a and b columns) should be compared with EE after cycle 5, which can be seen it has increased approximately three and six times after regenerations a and b, respectively.

## 3.5. Mutual solubility

Besides losing the materials, i.e. fuel and extractant, which can occur in any process, the mutual solubility of extractant and fuel change the composition and quality of the fuel. Therefore, the mutual solubility of TTD and fuel was investigated in particular to evaluate the solubility of this extractant in fuel because TTD is an N-containing extractant which its dissolution in oil will cause second contamination. For this purpose, the mutual solubility of pure heptane and TTD was checked as described in the experimental section. The UV/Vis spectrum of heptane after mixing with TTD and phase separation was recorded while pure heptane was used as the reference sample. The result (figure 6a) represents the appearance of a peak which indicates the existence of TTD as an impurity in heptane. Based on

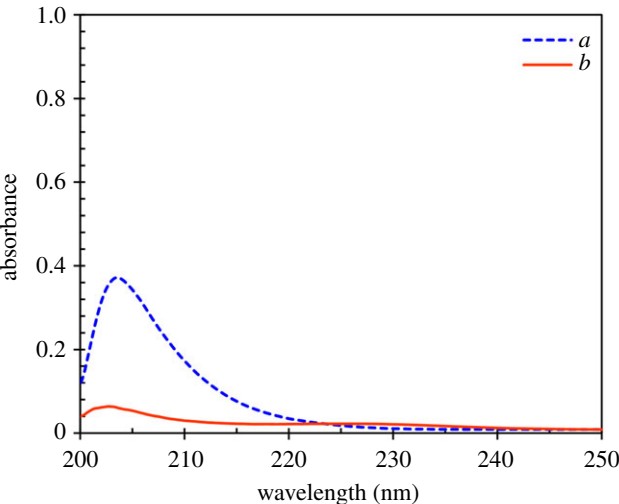

**Figure 6.** The UV/Vis spectrum of heptane after mixing with TTD and phase separation: before (*a*) and after (*b*) washing with water.

the extractant structure, it is expected that TTD can be extracted from the fuel by water washing. In the following, the UV/Vis spectrum of heptane after mixing with TTD, phase separation and finally water washing was recorded while pure heptane was used as the reference sample (figure 6*b*). As can be seen, the peak intensity greatly decreased after washing heptane with water. Therefore, it is concluded that the residual TTD in the fuel would not be the concern. Also, weighting the TTD phase, based on the description in the experimental section, showed the solubility of heptane in the extractant is approximately 5 wt%. The same or higher values was reported in the case of ionic liquids containing N atom [2,27].

## 3.6. Extraction mechanism

Gao *et al.* [8] showed that the interaction type of polyethers to DBT is the H-bonding between aromatic hydrogens of DBT and active heteroatoms of extractant. The formation of this type of interaction can be experimentally confirmed by comparing the $^1$H-NMR of sole DBT and TTD containing DBT [8,28]. There are three types of aromatic hydrogen in the structure of DBT which are assigned as a, b and c in figure 7. As can be seen from figure 7, peaks of all of these aromatic H atoms have moved to higher field with the participation of TTD. This observation implies increasing the shielding effect for them due to the enhancement of their electron density. Therefore, it can be concluded that these aromatic H atoms act as the H-bonding donors and the interaction of DBT with TTD is the H-bonding between active heteroatoms O and N of TTD and aromatic H atoms of DBT. But the determination of the existence of other type of the interaction and the contribution of each type of heteroatom in the H-bonding formation require more detailed investigation using theoretical and other experimental methods.

## 3.7. Comparison between 4,7,10-trioxatridecane-1,13-diamine and other extractants

The advantages of TTD for EDS can be highlighted from different viewpoints. In comparison with traditional nitrogen-containing molecular solvents such as DMF and so on [13] which are volatile, TTD showed good EDS performance the same as observed in the case of earlier-mentioned solvents, but this extractant has a very low vapour pressure. Therefore, this solvent can be considered as a green solvent for EDS [8]. From the other viewpoint, the obtained $K_N$ value of TTD for removal of DBT is similar to that of reported values of Lewis acid-containing ionic liquids such as [Bmim][AlCl$_4$] (approx. 4.09) and NMP-SnCl$_2$ (approx. 4.06) [8]. This is while the high viscosity and cost have challenged the practical application of ionic liquids in EDS [10]. It should be noted that TTD is a commercially available solvent used in different industries. Finally, as discussed in §3.1, this diamino-terminated PEG-based solvent is a more efficient extractant than PEG for EDS. In conclusion, TTD can be introduced as a commercially available molecular solvent, which has very low vapour pressure, for green and efficient EDS. It should be noted that investigations on the extraction mechanism and clarification of the role of amine and ether functional groups in the extraction of benzothiophene compounds by TTD are experimentally and theoretically in progress in our research group.

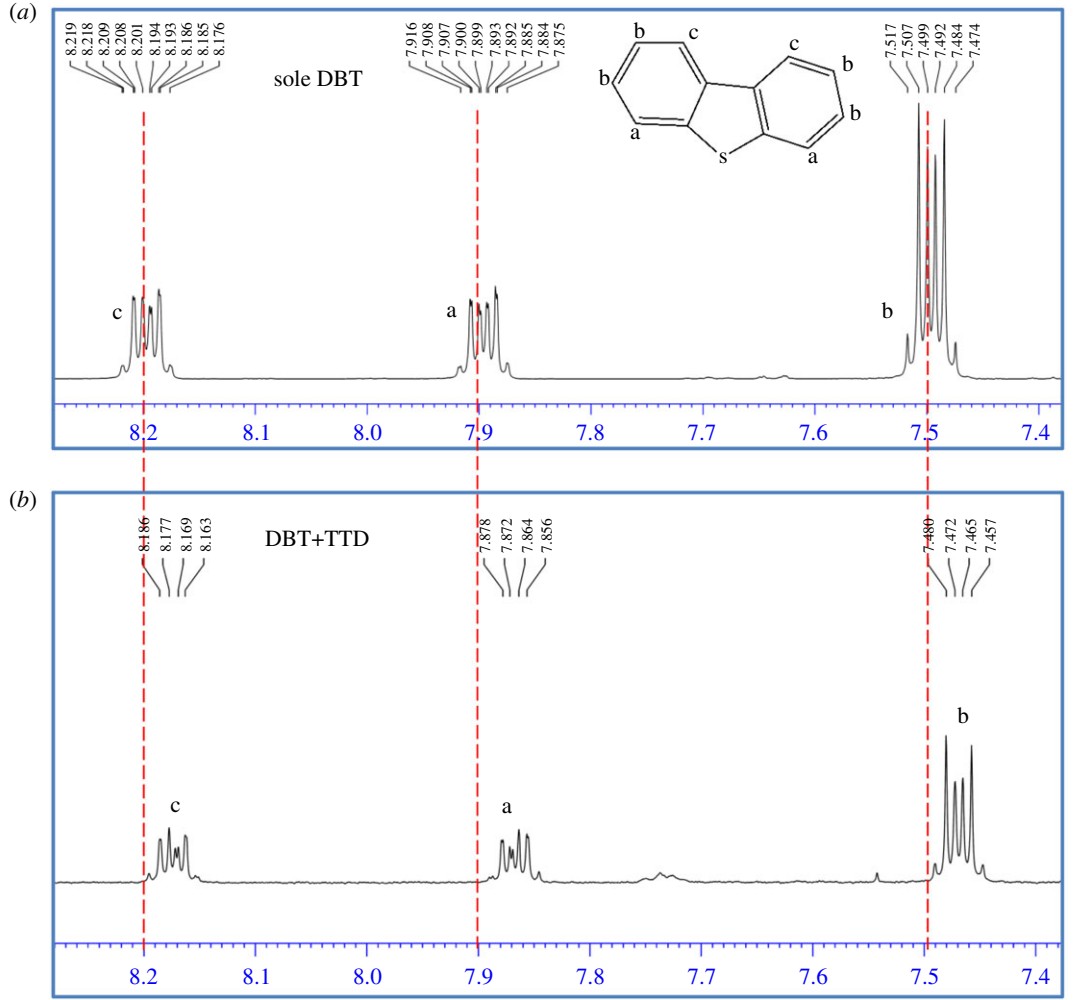

**Figure 7.** $^1$H-NMR spectra of (*a*) sole DBT and (*b*) its complex with TTD (DBT + TTD).

## 4. Conclusion

This work reports the performance of TTD, a diamine-terminated PEG-based molecular solvent with very low vapour pressure, for the EDS for the first time. The results showed that the TTD/fuel volume ratio of 0.5 could extract BT, DBT and DMDBT with the efficiencies 67%, 74% and 53%, respectively, in less than 1 min at ambient temperature. The obtained value of $K_N$ for removal of DBT by TTD (3.66 in (ppm S)$_{TTD}$/(ppm S)$_{fuel}$) was higher than that of PEG and it is similar to that of reported $K_N$ values about modified PEG and Lewis acid-containing ionic liquids. It was also observed that the deep EDS was achievable after three times the extraction using fresh TTD. It was observed that spent TTD after five cycles could be regenerated using the back-extraction method. Investigating the mutual solubility of fuel and extractant showed that the residual TTD in the fuel would not be the concerns because of removing them by water washing of fuel. Also, the result showed that the solubility of heptane in the extractant is approximately 5 wt%. Finally, study of the extraction mechanism using $^1$H-NMR showed that the interaction type of TTD to DBT is the H-bonding between aromatic hydrogens of DBT and active heteroatoms of extractant. These observations, as well as very low vapour pressure and insignificant dependency of TTD on the initial S-concentration of fuel and temperature, make this extractant to be introduced as a commercially available molecular solvent, which has very low vapour pressure, for green and efficient EDS.

Data accessibility. Our data are deposited at the Dryad Digital Repository: https://dx.doi.org/10.5061/dryad.931zcrjh1 [29].

Authors' contributions. F.R.M. carried out the experiment and analysis of the results; E.K. supervised the experiments and also wrote the manuscript; M.A.Z. and M.Y. conceived the original idea and provided materials; S.A. supervised the project.

Competing interests. We declare we have no competing interests.

Funding. We received no funding for this study

Acknowledgements. The authors acknowledge University of Mazandaran and Bu-Ali Sina University for support of this work.

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
