## [Reviewer comments · Royal Society Open Science]

Review History

RSOS-200803.R0 (Original submission)

Review form: Reviewer 1

Is the manuscript scientifically sound in its present form?

Yes

Are the interpretations and conclusions justified by the results?

Yes

Is the language acceptable?

Yes

Do you have any ethical concerns with this paper?

No

Have you any concerns about statistical analyses in this paper?

No

Recommendation?

Major revision is needed (please make suggestions in comments)

Comments to the Author(s)

Extractive desulfurization is a feasible technology in gasoline or diesel refining. The results from manuscript suggests that extractive desulfurization using TTD is feasible. However, the manuscript still needs major revisions. Some experiments should be supplemented to clarify what I am concerned about.

1. Regeneration of TTD is very important. So the authors should provide data about the regeneration.
2. Selectivity of sulfurs to aromatics is also important. The author should give the corresponding data regarding the selectivity.
3. How about the mutual solubility for TTD and model oil? TDD is N-containing extarctant. So its dissolution in oil will cause second contamination besides the extractant loss. Besides, the oil, especially the aromatic-containing oil, will be dissolved in TTD. This results in the oil loss. I do not think that TTD and the oil are immiscible completely.
4. Compared with PEGs and NHD, what are advantages of TTD?

Review form: Reviewer 2

Is the manuscript scientifically sound in its present form?

No

Are the interpretations and conclusions justified by the results?

No

Is the language acceptable?

No

Do you have any ethical concerns with this paper?

No

Have you any concerns about statistical analyses in this paper?

No

Recommendation?

Reject

Comments to the Author(s)

Fatemeh Rafiei Moghadam et al. studied extraction of benzothiophene, dibenzothiophene and dimethyl dibenzothiophene using diamine-terminated polyethylene glycol as potential green molecular solvent in desulfurization of liquid fuel. For that purpose they investigated the effects of extractant/fuel volume ratio, process time, temperature, initial S-compound concentration, S-compound type, multiple extraction and the recyclability of extractant on EDS.

Here are the main objections:

- 1) In Results and Discussion section, the authors claim to have investigated the recyclability of the extractant. Nevertheless, only the reuse has been investigated. The full potential of the recyclability can be achieved with the regeneration of the extractant only, and the authors did not investigate or at least offer such a possibility. I propose to examine all the regeneration possibilities, experimentally if possible.

2) The experiments are not presented in sufficient details. For example, the initial S-compound concentrations are not listed clearly and as a consequence, the experiments cannot be reproduced by the interested reader.

3) The overall amount of experiments is rather low and does not approve the publication. The full paper should make a greater contribution to the development of green technologies in the purification of liquid fuels. I propose to include the results of investigations on the extraction mechanism and clarification of the role of amine and ether functional groups in the extraction of benzothiophene compounds by TTD, which the authors suggest themselves in section 3.5, for future submission of the article.

The necessary corrections are listed below:

1. Although the authors used a word template file to prepare their article, the recommended sections were not followed. Results and Discussion sections should be written separately.
2. Summary section contains a reference which is not in accordance with the journal's recommendations.
3. Briefly explain what a molecular solvent is.
4. In the Introduction, the authors state a specific value of the vapor pressure of TTD (0.001 mmHg) and compare it as 'less than the PEG value'. This fact is confusing because they previously stated that the PEG value was 'less than 0.01 mmHg', which can generally mean that it is also less than the value of TTD. It is necessary to specifically define the value of PEG, as precisely as possible.
5. The experimental part (Section 2.2) is written too generally, and therefore is not entirely clear to understand. Modification of experimental description is necessary so that each reader can repeat the experiment from the description. It is necessary to clearly write the investigated S-compounds concentrations, the used quantities of chemicals and operating conditions - temperature, experiment time, volume ratio of TTD / fuel...
6. From the above mentioned, it is necessary to define why, how and for what purpose these experimental conditions were chosen.
7. From the text it is not clear that heptane is a model fuel - it has to be explicitly stated. Please explain why heptane was chosen as the model fuel. Moreover, BT, DBT and DM-DBT are typical S-components of diesel, and heptane is unfortunately far from being the best choice for one-component model diesel fuel. The boiling point of heptane is 98.42 °C, and boiling points of BT, DBT and DM-DBT are 221, 332.6 and 364.9 °C, respectively. Therefore, those compounds are unlikely to be present in the same refinery fraction. For instance, n-hexadecane with boiling point of 286.9 °C (similar to investigated S-compounds) would be a better choice.
8. In Section 3.3., a reference should be given to compare the results with PEG. (line 28-29)
9. In Figure 2, 4 i 5, the space is missing between 'of' and 'DBT' on y-axis.
10. In Figure 5 it is not clear which y-data refer to which x-data - the x-axis label needs to be adjusted.
11. The results quoted as Figure 5 are discussed as extraction efficiency (EE) within the text. However, the figure shows the dependence of DBT concentration on the number of extraction degrees. Although the quantities are mutually related, this way of presenting and discussing the results makes no sense.

Decision letter (RSOS-200803.R0)

Dear Dr Kianpour:

Title: Extractive desulfurization of liquid fuel using diamine-terminated polyethylene glycol as a very low vapor pressure and green molecular solvent
Manuscript ID: RSOS-200803

The editor assigned to your manuscript has now received comments from reviewers. We would like you to revise your paper in accordance with the referee and Subject Editor suggestions which can be found below (not including confidential reports to the Editor). Please note this decision does not guarantee eventual acceptance.

Please submit your revised paper before 27-Aug-2020. Please note that the revision deadline will expire at 00.00am on this date. If we do not hear from you within this time then it will be assumed that the paper has been withdrawn. In exceptional circumstances, extensions may be possible if agreed with the Editorial Office in advance. We do not allow multiple rounds of revision so we urge you to make every effort to fully address all of the comments at this stage. If deemed necessary by the Editors, your manuscript will be sent back to one or more of the original reviewers for assessment. If the original reviewers are not available we may invite new reviewers.

On behalf of the Subject Editor Professor Anthony Stace and the Associate Editor Dr Darren Walsh.

RSC Associate Editor:
Comments to the Author:
(There are no comments.)

RSC Subject Editor:
 Comments to the Author:
 (There are no comments.)

Reviewers' Comments to Author:
 Reviewer: 1

Comments to the Author(s)

Extractive desulfurization is a feasible technology in gasoline or diesel refining. The results from manuscript suggests that extractive desulfurization using TTD is feasible. However, the manuscript still needs major revisions. Some experiments should be supplemented to clarify what I am concerned about.

1. Regeneration of TTD is very important. So the authors should provide data about the regeneration.
2. Selectivity of sulfurs to aromatics is also important. The author should give the corresponding data regarding the selectivity.
3. How about the mutual solubility for TTD and model oil? TDD is N-containing extarctant. So its dissolution in oil will cause second contamination besides the extractant loss. Besides, the oil, especially the aromatic-containing oil, will be dissolved in TTD. This results in the oil loss. I do not think that TTD and the oil are immiscible completely.
4. Compared with PEGs and NHD, what are advantages of TTD?

Reviewer: 2

Comments to the Author(s)

Fatemeh Rafiei Moghadam et al. studied extraction of benzothiophene, dibenzothiophene and dimethyl dibenzothiophene using diamine-terminated polyethylene glycol as potential green molecular solvent in desulfurization of liquid fuel. For that purpose they investigated the effects of extractant/fuel volume ratio, process time, temperature, initial S-compound concentration, S-compound type, multiple extraction and the recyclability of extractant on EDS.

Here are the main objections:

- 1) In Results and Discussion section, the authors claim to have investigated the recyclability of the extractant. Nevertheless, only the reuse has been investigated. The full potential of the recyclability can be achieved with the regeneration of the extractant only, and the authors did not investigate or at least offer such a possibility. I propose to examine all the regeneration possibilities, experimentally if possible.
- 2) The experiments are not presented in sufficient details. For example, the initial S-compound concentrations are not listed clearly and as a consequence, the experiments cannot be reproduced by the interested reader.
- 3) The overall amount of experiments is rather low and does not approve the publication. The full paper should make a greater contribution to the development of green technologies in the purification of liquid fuels. I propose to include the results of investigations on the extraction mechanism and clarification of the role of amine and ether functional groups in the extraction of benzothiophene compounds by TTD, which the authors suggest themselves in section 3.5, for future submission of the article.

The necessary corrections are listed below:

1. Although the authors used a word template file to prepare their article, the recommended sections were not followed. Results and Discussion sections should be written separately.
2. Summary section contains a reference which is not in accordance with the journal's recommendations.
3. Briefly explain what a molecular solvent is.

4. In the Introduction, the authors state a specific value of the vapor pressure of TTD (0.001 mmHg) and compare it as 'less than the PEG value'. This fact is confusing because they previously stated that the PEG value was 'less than 0.01 mmHg', which can generally mean that it is also less than the value of TTD. It is necessary to specifically define the value of PEG, as precisely as possible.
5. The experimental part (Section 2.2) is written too generally, and therefore is not entirely clear to understand. Modification of experimental description is necessary so that each reader can repeat the experiment from the description. It is necessary to clearly write the investigated S-compounds concentrations, the used quantities of chemicals and operating conditions – temperature, experiment time, volume ratio of TTD / fuel...
6. From the above mentioned, it is necessary to define why, how and for what purpose these experimental conditions were chosen.
7. From the text it is not clear that heptane is a model fuel – it has to be explicitly stated. Please explain why heptane was chosen as the model fuel. Moreover, BT, DBT and DM-DBT are typical S-components of diesel, and heptane is unfortunately far from being the best choice for one-component model diesel fuel. The boiling point of heptane is 98.42 °C, and boiling points of BT, DBT and DM-DBT are 221, 332.6 and 364.9 °C, respectively. Therefore, those compounds are unlikely to be present in the same refinery fraction. For instance, n-hexadecane with boiling point of 286.9 °C (similar to investigated S-compounds) would be a better choice.
8. In Section 3.3., a reference should be given to compare the results with PEG. (line 28-29)
9. In Figure 2, 4 i 5, the space is missing between 'of' and 'DBT' on y-axis.
10. In Figure 5 it is not clear which y-data refer to which x-data – the x-axis label needs to be adjusted.
11. The results quoted as Figure 5 are discussed as extraction efficiency (EE) within the text. However, the figure shows the dependence of DBT concentration on the number of extraction degrees. Although the quantities are mutually related, this way of presenting and discussing the results makes no sense.

Author's Response to Decision Letter for (RSOS-200803.R0)

See Appendix A.

RSOS-200803.R1 (Revision)

Review form: Reviewer 1

Is the manuscript scientifically sound in its present form?

Yes

Are the interpretations and conclusions justified by the results?

Yes

Is the language acceptable?

Yes

Do you have any ethical concerns with this paper?

No

Have you any concerns about statistical analyses in this paper?

No

Recommendation?

Accept with minor revision (please list in comments)

Comments to the Author(s)

1. As shown in Fig. 5, extraction efficiency of TTD greatly reduces after solvent washing. So, the authors should further explain what causes this significant reduction.
2. Fig. 6 provides UV spectrum for TTD in heptane. However, there's no specific data of the solubility of TTD in heptane. As far as I know, UV can give this data.

Review form: Reviewer 2

Is the manuscript scientifically sound in its present form?

No

Are the interpretations and conclusions justified by the results?

Yes

Is the language acceptable?

No

Do you have any ethical concerns with this paper?

No

Have you any concerns about statistical analyses in this paper?

No

Recommendation?

Accept with minor revision (please list in comments)

Comments to the Author(s)

Comments to the Author(s) after revision

Most of the suggestions have been corrected, but the manuscript still needs minor revisions.

- The authors need to complete investigations on the extraction mechanism and include them in this manuscript. The mutual solubility and regeneration of TTD are basic experiments in this research area, and they can not significantly improve the quality of the work for publication.
- The author's answer(3) about molecular solvent should be included in the manuscript.
- The author's answer(6) about experimental conditions choice should be included in the manuscript.

Decision letter (RSOS-200803.R1)

Dear Dr Kianpour:

Title: Extractive desulfurization of liquid fuel using diamine-terminated polyethylene glycol as a very low vapor pressure and green molecular solvent
Manuscript ID: RSOS-200803.R1

Thank you for submitting the above manuscript to Royal Society Open Science. On behalf of the Editors and the Royal Society of Chemistry, I am pleased to inform you that your manuscript will be accepted for publication in Royal Society Open Science subject to minor revision in accordance with the referee suggestions. Please find the reviewers' comments at the end of this email.

The reviewers and handling editors have recommended publication, but also suggest some minor revisions to your manuscript. Therefore, I invite you to respond to the comments and revise your manuscript.

Because the schedule for publication is very tight, it is a condition of publication that you submit the revised version of your manuscript before 01-Oct-2020. Please note that the revision deadline will expire at 00.00am on this date. If you do not think you will be able to meet this date please let me know immediately.

Once again, thank you for submitting your manuscript to Royal Society Open Science. The chemistry content of Royal Society Open Science is published in collaboration with the Royal

Society of Chemistry. I look forward to receiving your revision. If you have any questions at all, please do not hesitate to get in touch.

Kind regards,
Dr Laura Smith
Publishing Editor, Journals

On behalf of the Subject Editor Professor Anthony Stace and the Associate Editor Dr Darren Walsh.

RSC Associate Editor:
Comments to the Author:
(There are no comments.)

RSC Subject Editor:
Comments to the Author:
(There are no comments.)

Reviewer comments to Author:
Reviewer: 1

Comments to the Author(s)

1. As shown in Fig. 5, extraction efficiency of TTD greatly reduces after solvent washing. So, the authors should further explain what causes this significant reduction.
2. Fig. 6 provides UV spectrum for TTD in heptane. However, there's no specific data of the solubility of TTD in heptane. As far as I know, UV can give this data.

Reviewer: 2

Comments to the Author(s)

Comments to the Author(s) after revision

Most of the suggestions have been corrected, but the manuscript still needs minor revisions.

- The authors need to complete investigations on the extraction mechanism and include them in this manuscript. The mutual solubility and regeneration of TTD are basic experiments in this research area, and they can not significantly improve the quality of the work for publication.
- The author's answer(3) about molecular solvent should be included in the manuscript.
- The author's answer(6) about experimental conditions choice should be included in the manuscript.

Author's Response to Decision Letter for (RSOS-200803.R1)

See Appendix B.

Decision letter (RSOS-200803.R2)

Dear Dr Kianpour:

Title: Extractive desulfurization of liquid fuel using diamine-terminated polyethylene glycol as a very low vapor pressure and green molecular solvent
Manuscript ID: RSOS-200803.R2

It is a pleasure to accept your manuscript in its current form for publication in Royal Society Open Science. The chemistry content of Royal Society Open Science is published in collaboration with the Royal Society of Chemistry.

On behalf of the Subject Editor Professor Anthony Stace and the Associate Editor Dr Darren Walsh.

RSC Associate Editor
Comments to the Author:
(There are no comments.)

Reviewer(s)' Comments to Author:

Appendix A

Response to the reviewers' comments:

Reviewer: 1

Comments to the Author(s)

Extractive desulfurization is a feasible technology in gasoline or diesel refining. The results from manuscript suggests that extractive desulfurization using TTD is feasible. However, the manuscript still needs major revisions. Some experiments should be supplemented to clarify what I am concerned about.

1. Regeneration of TTD is very important. So the authors should provide data about the regeneration.

Authors reply: Based on your suggestion, regeneration of extractant was studied. Results are discussed and highlighted in the manuscript and the obtained data are presented in Figure 5.

2. Selectivity of sulfurs to aromatics is also important. The author should give the corresponding data regarding the selectivity.

Authors reply: You are right but it is not possible for us to investigate selectivity of sulfurs to aromatics because of, absence of required apparatus in our laboratory and the limitation of using other places caused due to COVID 19.

3. How about the mutual solubility for TTD and model oil? TDD is N-containing extractant. So its dissolution in oil will cause second contamination besides the extractant loss. Besides, the oil, especially the aromatic-containing oil, will be dissolved in TTD. This results in the oil loss. I do not think that TTD and the oil are immiscible completely.

Authors reply: The mutual solubility of TTD and model oil was studied. Results are discussed and highlighted in the manuscript and obtained data are presented in Figure 6.

4. Compared with PEGs and NHD, what are advantages of TTD?

Authors reply: Results of this work showed that the TTD/fuel volume ratio of 0.5 which is equal to mass ratio 0.74 could extract DBT with the efficiency 74% and the distribution coefficient (K_N) value 3.66. Therefore, TTD is more efficient extractant than PEGs (200, 400 and 600). Because the extractant/fuel mass ratio 1/1 of PEGs

extracted 75% of DBT with the average value of 2.2 for K_N . [8]. Moreover, TTD, with vapor pressure <0.001 hPa, has lower pressure than PEGs (<0.01 hPa) as well as NHD (<0.01 hPa). In the case of extractive desulfurization performance, almost the same results were obtained for TTD and NHD.

Reviewer: 2

Comments to the Author(s)

Fatemeh Rafiei Moghadam et al. studied extraction of benzothiophene, dibenzothiophene and dimethyl dibenzothiophene using diamine-terminated polyethylene glycol as potential green molecular solvent in desulfurization of liquid fuel. For that purpose they investigated the effects of extractant/fuel volume ratio, process time, temperature, initial S-compound concentration, S-compound type, multiple extraction and the recyclability of extractant on EDS.

Here are the main objections:

1) In Results and Discussion section, the authors claim to have investigated the recyclability of the extractant. Nevertheless, only the reuse has been investigated. The full potential of the recyclability can be achieved with the regeneration of the extractant only, and the authors did not investigate or at least offer such a possibility. I propose to examine all the regeneration possibilities, experimentally if possible.

Authors reply: Based on your suggestion, regeneration of extractant was studied. Results are discussed and highlighted in the manuscript and obtained data are presented in Figure 5.

2) The experiments are not presented in sufficient details. For example, the initial S-compound concentrations are not listed clearly and as a consequence, the experiments cannot be reproduced by the interested reader.

Authors reply: To provide the sufficient details, the experimental section was rewritten and the changes are highlighted in the manuscript.

3) The overall amount of experiments is rather low and does not approve the publication. The full paper should make a greater contribution to the development of green technologies in the purification of liquid fuels. I propose to include the results of investigations on the extraction mechanism and clarification of the role of amine and

ether functional groups in the extraction of benzothiophene compounds by TTD, which the authors suggest themselves in section 3.5, for future submission of the article.

Authors reply: To increase the amount of experiments and also improve the quality of the work, the mutual solubility as well as regeneration of TTD were investigated. About the extraction mechanism, complete and definitive results have not been obtained yet.

The necessary corrections are listed below:

1. Although the authors used a word template file to prepare their article, the recommended sections were not followed. Results and Discussion sections should be written separately.

Authors reply: This format is usual in the “*Royal Society Open Science*.”. Some examples are:

1- Ethylene glycol assisted three-dimensional floral evolution of BiFeO₃-based nanostructures with effective magneto-electric response (Published:05 August 2020, Article ID:200642).

2- Combining derivative and synchronous approaches for simultaneous spectrofluorimetric determination of terbinafine and itraconazole (Published:05 August 2020, Article ID:200571)

2. Summary section contains a reference which is not in accordance with the journal's recommendations.

Authors reply: It was corrected.

3. Briefly explain what a molecular solvent is.

Authors reply: A molecular solvent means an ordinary liquid such as water, chloroform and etc. which are predominantly made of electrically neutral molecules (non-polar or polar molecules but not ionic molecules). This term is sometimes used in scientific texts to distinguish these classical solvents from ionic liquids, emerging solvents which are largely made of ions and short-lived ion pairs. For example: Comparing an Ionic Liquid to a Molecular Solvent in the Cesium Cation Extraction by a Calixarene: A Molecular Dynamics Study of the Aqueous Interfaces (J. Phys. Chem. B 2006, 110, 39, 19497–19506).

4. In the Introduction, the authors state a specific value of the vapor pressure of TTD (0.001 mmHg) and compare it as ‘less than the PEG value’. This fact is confusing

because they previously stated that the PEG value was ‘less than 0.01 mmHg’, which can generally mean that it is also less than the value of TTD. It is necessary to specifically define the value of PEG, as precisely as possible.

Authors reply: Thank you for careful scrutiny of the manuscript. Based on the available data these values were corrected as “PEG-200 with the vapor pressure <0.01 hPa” and “TTD has a very low vapor pressure (<0.001 hPa...)”.

5. The experimental part (Section 2.2) is written too generally, and therefore is not entirely clear to understand. Modification of experimental description is necessary so that each reader can repeat the experiment from the description. It is necessary to clearly write the investigated S-compounds concentrations, the used quantities of chemicals and operating conditions – temperature, experiment time, volume ratio of TTD / fuel...

Authors reply: The experimental conditions of each test are presented in the figure captions, however this section was rewritten based on your suggestions and changes were highlighted in the manuscript.

6. From the above mentioned, it is necessary to define why, how and for what purpose these experimental conditions were chosen.

Authors reply: About temperature, as mentioned in introduction, extractive desulfurization has received special attention due to its convenient operating conditions including ambient temperature. Therefore taking into account 10 degree fluctuations in ambient temperature, temperature range 15-35 °C was chosen to investigate the effect of this parameters on the EDS. Because of very low temperature dependence of EDS using TTD, room temperature was chosen as optimum condition for process temperature.

About the volume ratio, as mentioned in section 3.1, this condition was selected based on obtained data during the optimization of EE vs. the volume ratio. Extraction time 20 min was considered to assure about reaching the equilibrium condition. In the case of sulfur concentration, influents with different sulfur concentrations are injected into the desulfurization units of refineries and these initial concentrations were chosen based on common concentrations investigated in this type of research. Some references are as follow.

1- Green Chem., 2004, 6, 316-322.

2- Fuel Process. Technol., 2014, 123, 1-10.

3- Chem. Eng. J., 2015, 274, 192-199

4- Scientific Reports, 2020, DOI: 10.1038/s41598-020-67235-8.

7. From the text it is not clear that heptane is a model fuel – it has to be explicitly stated. Please explain why heptane was chosen as the model fuel. Moreover, BT, DBT and DM-DBT are typical S-components of diesel, and heptane is unfortunately far from being the best choice for one-component model diesel fuel. The boiling point of heptane is 98.42 °C, and boiling points of BT, DBT and DM-DBT are 221, 332.6 and 364.9 °C, respectively. Therefore, those compounds are unlikely to be present in the same refinery fraction. For instance, n-hexadecane with boiling point of 286.9 °C (similar to investigated S-compounds) would be a better choice.

Authors reply: It is noted in the first line of section 2.2 that heptane is used to prepare the model fuel “In order to prepare the model liquid fuel ... was dissolved in heptane”. About the second part of this question, you are right but a literature survey shows that many works (a few are listed below: Ref. 1-4) used n-heptane, n-octane or even n-hexane to prepare a model fuel containing mentioned benzothiophenic compounds. Based on what is explained in the following, it can be concluded that type of alkane used to prepare the model liquid fuel has little effect on results. The basis of the extractive desulfurization is dissolution and distribution of sulfur compound between fuel and extractant. On the other hand the Hildebrand solubility parameter (δ) provides an estimate of the degree of interaction between materials and can be a good indication of solubility so that the smaller the difference between Hildebrand solubility parameters of solvent and solute, the more solubility. The Hildebrand solubility parameter values of n-hexane, n-heptane, n-octane and n-hexadecane are equal 14.9, 15.3, 15.5 and 16.3 (Ref. 5-7), respectively. As it can be seen there are little differences between the Hildebrand solubility parameter values of these alkanes. Therefore, it can be concluded that type of these alkanes, used to prepare the model liquid fuel, has little effect on results and many works used n-heptane or n-octane to prepare a model fuel containing mentioned benzothiophenic compounds.

1- Fuel, 2018, 233, 704-13.

2- Energy Fuels, 2012, 26, 3723-3727.

3- Journal of Environmental Chemical Engineering, 2020, 8, 104182.

4- Fuel, 2008, 87, 79-84.

5- <https://cool.culturalheritage.org/byauth/burke/solpar/solpar2.html>

6- https://www.accudynetest.com/solubility_table.html#004

7- <https://cool.culturalheritage.org/coolaic/sg/bpg/annual/v03/bp03-04.html>

8. In Section 3.3., a reference should be given to compare the results with PEG. (line 28-29)

Authors reply: The reference was inserted.

9. In Figure 2, 4 i 5, the space is missing between ‘of’ and ‘DBT’ on y-axis.

Authors reply: Thanks for your careful scrutiny. They were corrected.

10. In Figure 5 it is not clear which y-data refer to which x-data – the x-axis label needs to be adjusted.

Authors reply: This Figure was represented.

11. The results quoted as Figure 5 are discussed as extraction efficiency (EE) within the text. However, the figure shows the dependence of DBT concentration on the number of extraction degrees. Although the quantities are mutually related, this way of presenting and discussing the results makes no sense.

Authors reply: Its explanation in the text was revised.

Appendix B

Response to the reviewers' comments:

Reviewer: 1

Comments to the Author(s)

1. As shown in Fig. 5, extraction efficiency of TTD greatly reduces after solvent washing. So, the authors should further explain what causes this significant reduction.

Authors reply: It is to be noted that the extraction ability of DBT by TTD after regeneration (reg. a and b columns) should be compared with EE after cycle 5, which can be seen it has increased about 3 and 6 times after regenerations a and b. These sentences were added and highlighted in manuscript to more clarify the discussion.

2. Fig. 6 provides UV spectrum for TTD in heptane. However, there's no specific data of the solubility of TTD in heptane. As far as I know, UV can give this data.

Authors reply: The aim of presentation of UV spectrum before and after washing with water is to show that it is possible to remove the solved TTD in heptane by water washing. So its solubility value is not important in the present study.

Reviewer: 2

Comments to the Author(s)

Most of the suggestions have been corrected, but the manuscript still needs minor revisions.

- The authors need to complete investigations on the extraction mechanism and include them in this manuscript. The mutual solubility and regeneration of TTD are basic experiments in this research area, and they cannot significantly improve the quality of the work for publication.

Authors reply: Based on your suggestion the extraction mechanism was investigated using ¹H-NMR study. Details of test, results (Fig. 7) and discussion were inserted and highlighted in the manuscript.

- The author's answer (3) about molecular solvent should be included in the manuscript.

Authors reply: It was done based on your suggestion and highlighted in the manuscript.

- The author's answer (6) about experimental conditions choice should be included in the manuscript.

Authors reply: It was done based on your suggestion and highlighted in the manuscript.

Finally, the authors acknowledge the reviewers, whose comments really improved the quality of our manuscript.